# A Scoping Review on the Economic Impacts of Healthy Ageing Promotion and Disease Prevention in OECD Member Countries

**DOI:** 10.3390/ijerph22081161

**Published:** 2025-07-22

**Authors:** Ezgi Dilek Demirtas, Antoine Flahault

**Affiliations:** 1Institute of Economic Research, Faculty of Economics and Business, University of Neuchâtel, Avenue du Premier-Mars 26, 2000 Neuchâtel, Switzerland; 2Institute of Global Health, Faculty of Medicine, University of Geneva, 9 Chemin des Mines, 1202 Geneva, Switzerland; antoine.flahault@unige.ch

**Keywords:** promotion, prevention, ageing, economics

## Abstract

The economic impact of health promotion and disease prevention interventions in ageing populations remains debated, as theories of morbidity compression and expansion offer contrasting views on the relationship between life expectancy and duration of morbidity. A MEDLINE search was conducted to identify studies evaluating the economic impact of health promotion or primary or secondary prevention interventions in OECD countries, over a lifetime time horizon. Among the 29 studies included, 16 reported cost-saving interventions (reducing costs while improving health outcomes), 11 reported cost-effective interventions (health gains at an acceptable additional cost based on an established threshold), and two presented cost-ineffective interventions (costs exceeding the threshold for the health benefits achieved). Interventions targeting diabetes and obesity prevention were cost-saving; cancer screening and fall prevention strategies were cost-effective; whereas interventions targeting rare diseases were cost-ineffective. Regulatory interventions were also cost-saving, while most programme-based interventions were cost-effective. Cost-saving or cost-effective interventions generally adopted broader analytical perspectives, while cost-ineffective ones employed narrower perspectives. The four studies that incorporated competing risks—despite using a narrower healthcare sector perspective—still found the interventions to be cost-saving or cost-effective interventions. None of the included studies assessed whether interventions led to morbidity compression or expansion. Only a few studies considered equity impact; those that did reported improved outcomes for disadvantaged groups, in regulatory and community-based interventions. Further research is needed to quantify morbidity outcomes and enhance methodological consistency, particularly with respect to analytical perspectives, the integration of competing risks, and the inclusion of equity analyses.

## 1. Introduction

In political environments where decision-makers prioritise interventions that yield visible outcomes within a single election cycle, investment in health promotion and primary or secondary disease prevention—whose benefits typically emerge over the long term—often receives limited political support. This short-term focus contributes to the underfunding of preventive measures, despite their evidence-based capacity to improve population health and their potential to better control healthcare costs over time. In 2015, OECD member countries allocated, on average, less than 3% of their healthcare spending to health promotion and prevention [1]. This share rose to 5% in 2021 [2]. However, this increase was largely driven by temporary COVID-19-related measures—such as testing, surveillance, and vaccination—rather than sustained, planned investments in population health. As such, it may not reflect a genuine shift toward long-term investment in health promotion and disease prevention.

The observed association between increased life expectancy and rising per capita healthcare expenditure has often led to ageing being blamed for escalating healthcare costs. However, the relationship between ageing and healthcare expenditure is complex and influenced by various factors, including the health status of older populations [3]. The effects of health promotion and disease prevention on population ageing remain subject to debate. Two contrasting theories—compression of morbidity and expansion of morbidity—offer different perspectives on this issue. The compression of morbidity theory argues that as life expectancy increases, the onset of disability is delayed, resulting in a shorter period of morbidity [4]. In contrast, the expansion of morbidity theory posits that while life expectancy increases, the onset of disability remains unchanged, leading to a longer period of ill health and higher healthcare costs [5]. These competing theories add further complexity to the challenge of funding promotion and prevention.

Numerous reviews of empirical studies have shown that health promotion and preventive measures are cost-saving (i.e., they reduce costs while maintaining or improving health outcomes) or cost-effective (i.e., they yield health benefits at an acceptable additional cost based on established thresholds), and they contribute to healthier ageing populations [6,7,8,9]. Van der Vliet et al. identified 51 promotion and prevention interventions from sectors beyond healthcare—including environmental, occupational, and behavioural domains—that were found to be cost-saving or cost-effective [10]. Despite the well-documented health benefits, promotion and prevention strategies remain underutilised. This is partly due to methodological challenges in economic evaluations and the difficulty of demonstrating long-term economic impacts. The lack of comprehensive data continues to hamper evidence-informed policymaking.

This paper aims to review published literature on the morbidity and net economic impacts of health promotion and disease prevention interventions. The objective of this scoping review is to assess the extent and nature of existing evidence, with a particular focus on studies using a lifetime time horizon to account for competing risks arising over the course of life. OECD member countries were included due to their ageing populations and the availability of comparable health and economic data.

A preliminary search of MEDLINE, the Cochrane Database of Systematic Reviews, and *JBI Evidence Synthesis* revealed no existing or ongoing systematic or scoping reviews on this specific topic. To our knowledge, this is the first scoping review to investigate the net economic impact of prevention and promotion interventions in the context of ageing populations in OECD member countries.

## 2. Materials and Methods

A scoping literature review was conducted. Ethical approval was not sought, as the research consisted solely of a review of publicly available information. The review was not registered with a protocol. The Preferred Reporting Items for Systematic Reviews and Meta-Analyses extension for Scoping Reviews (PRISMA-ScR) Checklist [11] was used to guide and ensure clear reporting of the review (Appendix A).

We selected all English and French language studies conducted in OECD member countries that reported quantitative data on the economic impact of health promotion or primary or secondary disease prevention while adopting a lifetime time horizon. The review focused exclusively on primary and secondary prevention interventions, as these aim to prevent disease onset or support early detection, thereby offering substantial potential for health gains and cost control. Tertiary prevention, which involves managing established disease, was excluded to maintain the focus on prevention. The intervention was not limited to an age group, but the studies had to quantify the intervention’s impact over the lifetime of participants. Studies without a control group were excluded. Failure to address the morbidity compression or expansion dimension was not an exclusion criterion. Nonetheless, such studies were classified as not addressing this dimension in the analysis. Eligible empirical studies were either experimental (randomised controlled trials, non-randomised controlled trials), or observational (cohort and case–control studies). Cross-sectional studies, case-series, case-reports, and quasi-experimental studies (before-and-after studies, interrupted time-series studies) were excluded, as they do not allow the adoption of a lifetime horizon or lack a control group. Publications not reporting primary data—such as opinion articles, commentaries, editorials, or methodology articles—were also excluded. Appendix A details the eligibility criteria.

A single reviewer (the first author) conducted the search in MEDLINE via Ovid. The search strategy was structured around three concepts: promotion and prevention, morbidity impact, and economic impact. Each concept combined free-text terms (keywords and synonyms) and Medical Subject Headings (MeSH), ensuring a comprehensive approach. Truncation, inclusion of words with alternative letters, absent letters, and proximity searching were used to improve sensitivity. Subject headings were “exploded” to include more specific related terms; otherwise, terms were selected individually. All subheadings were included. For example, the following terms were used to search for health outcomes related to promotion and prevention: keyword search—“(healthy ag#ing) or (morbidity adj1 compression) or (delayed morbidity) or (pure ag#ing) or (morbidity adj1 expansion)”—and subject heading search—“aging/or Healthy Aging/or Morbidity/or exp Health Status/”. Boolean operators (AND/OR) were applied to combine results. Filters were used to select only studies published in English or French. The complete search strategy is presented in Appendix A. Only studies published from 2016 to the date of the search (8 May 2025) were included. This timeframe was chosen to capture recent, policy-relevant evidence reflecting contemporary healthcare practices and economic contexts. Screening, data extraction, and synthesis of studies published between 2016 and July 2024 were conducted between July 2024 and December 2024. To ensure the review was up to date at the time of submission, the search was rerun on 8 May 2025, to identify additional studies published between July 2024 and May 2025. In response to a reviewer’s comment, four additional studies not identified through the search strategy—but meeting the eligibility criteria—were included.

A standardised data extraction form was developed, capturing the following variables: country, intervention, comparison, population, intervention period, perspective, methods, discounting, sensitivity analyses methods, summary of results, morbidity impact, competing risks analysis, equity analysis, detailed results, and sensitivity analyses results. Data extraction was performed by a single reviewer (the first author). When authors did not specify the analytical perspective used for outcome and cost evaluation, the classification by Sittimart et al.’s [12] was applied. Since all relevant data were accessible from the publications, no authors were contacted for missing or additional data. In line with OECD guidelines, all cost data were adjusted to 2022 values using country-specific Consumer Price Indices and converted to USD 2022 using OECD Purchasing Power Parities to ensure comparability across countries and over time.

To assess whether studies examined the intervention’s morbidity impact, we applied predefined criteria during data extraction. Specifically, we looked for outcomes or statements indicating whether the period lived with morbidity was shortened or prolonged, or whether life expectancy gains were accompanied by less or more than proportional changes in measures that incorporate morbidity (e.g., Health-Adjusted Life Expectancy, Quality-Adjusted Life Expectancy, Disability-Free Life Expectancy, Healthy Life Years, Years of Healthy Life). Studies that reported changes in total morbidity burden (e.g., QALYs or DALYs or life years gained) without specifying the duration or proportion of morbidity relative to life expectancy were considered not to have addressed the morbidity compression or expansion dimension.

Economic impacts were classified as cost-saving, cost-effective, or cost-ineffective based on the cost-effectiveness threshold reported by the study.

Articles were quality-assessed using the Drummond 10-item checklist for economic evaluations [13].

## 3. Results

We identified 1799 records through our database search and by incorporating additional studies recommended by a reviewer, after the removal of duplicates. All citations were imported into the bibliographic software EndNote 20. One reviewer conducted the screening process. Titles and abstracts were assessed against the eligibility criteria, and the reasons for exclusion were recorded. Potentially relevant shortlisted studies were retrieved in full text. Full-text articles were retrieved for all studies deemed potentially relevant and subsequently assessed for eligibility. The search results and study selection process are summarised in a PRISMA flow diagram (see Figure 1).

### 3.1. Overview of Included Studies

Most studies were conducted in the United States (*n* = 8) [14,15,16,17,18,19,20,21], the United Kingdom (*n* = 8) [22,23,24,25,26,27,28,29], or Australia (*n* = 4) [30,31,32,33]). All studies were cost-effectiveness analyses based on empirical data and employed a decision–analytic modelling approach. Of the 29 included studies, 16 reported cost-saving promotion or prevention interventions. Eleven studies found the interventions to be cost-effective, while only two studies reported interventions that were not cost-effective. Table 1 summarises this review’s findings per economic impact category (cost-saving, cost-effective and not cost-effective), detailing each study’s the of intervention, study country, target population, adopted perspective, inclusion of competing risks, and whether an equity analysis was conducted. Unless otherwise stated, the following section presents the main findings of the included studies. Additional information, including the intervention calendar period, methods, discounting, detailed results, and sensitivity analyses methods and results are available in Appendix A.

### 3.2. Economic Impact Categories

#### 3.2.1. Cost-Saving Interventions (16 Studies)

Applying a sugar-sweetened beverage (SSB) tax in Canada led to a gain of 1.5 million Quality-Adjusted Life Years (QALYs) (95% UI 1.3 to 1.7 million) and saved USD 36.14 billion (95% UI 32.86 to 38.29 billion) over a lifetime horizon from the public healthcare sector perspective. The incremental cost-effectiveness ratio (ICER) was USD −23,987 QALY (95% UI −26,545 to −22,177/QALY). From a societal perspective, the tax was projected to generate USD 42.38 billion in tax revenue over 80 years [34].

Lifestyle modification and metformin administration for pre-diabetic individuals in the United States (US) were compared to usual care, with interventions applied either to all participants or targeted to those in the highest diabetes risk quintile. Lifestyle modification produced health benefits and reduced lifetime costs for all individuals, with net monetary benefits (NMB) ranging from USD 9124 (SD 4638) to 73,756 (23,207). Metformin showed more variable results, with NMBs ranging from USD −5423 (SD 4296) to 40,139 (11,773), depending on risk level and the assumed duration of treatment effects. Avoiding metformin use in lower-risk groups, when costs outweighed benefits, resulted in a NMB of USD 646 to 1088/person. Targeting metformin to the highest risk quintile improved cost-effectiveness, yielding NMB ranging from USD 19,778 to 24,722/person [14].

Menu calorie labelling among US adults was found to be cost-saving based on consumer response alone, resulting in net lifetime savings of USD 12.12 billion from the healthcare sector perspective and USD 14.80 billion from a societal perspective after accounting for implementation and compliance costs. Industry reformulation, such as a 5% calorie reduction, was projected to further increase savings to USD 16.44 billion and USD 21.79 billion from healthcare and societal perspectives, respectively [15]. These figures represent 0.27 to 0.49% of total health expenditure in the US in 2019 [43].

Fifteen obesity prevention interventions evaluated in an Australian cohort model included both regulatory and programme-based approaches. Of these, eleven were cost-saving, and four were cost-effective. Regulatory interventions generally ranked higher in cost-effectiveness. Among the cost-saving interventions, 11 were regulatory—such as alcohol tax and SSB taxes, portion size caps, advertising restrictions, menu labelling—and four were programme-based, including media campaigns, shelf tags on healthier products, and school-based interventions [31].

A SSB size cap modelled in a New Zealand cohort was estimated to yield 82,100 QALYs (95% UI 65,100 to 101,000) and USD 1.44 billion (95% UI 1.02 to 1.87 billion) in healthcare cost savings, assuming no compensation in energy intake. Scenario analyses with 20 to 100% energy compensation still resulted in cost savings ranging from USD 1.23 to 0.36 billion. The persisting health and cost benefits, even with full energy compensation, were hypothesised to result from the direct impact of SSBs on diabetes risk, independent of Body Mass Index [35].

Size cap and kilojoule reduction in SSBs modelled in the Australian population resulted in an estimated gain of 218,454 (95% UI 166,088 to 286,112) Health-Adjusted Life Years (HALYs) and cost savings of USD 2.07 billion (95% UI 1.13 to 2.35 billion) from a limited societal perspective [33].

From a healthcare sector perspective in Australia, and using a model that accounted for competing risks, a combined package of taxes on saturated fat, salt, sugar, and SSBs, along with subsidies on fruits and vegetables, was cost-saving and could avert up to 470,000 DALYs (95% UI 420,000 to 510,000) with net savings of USD 3.16 billion (95% UI USD 2.2 to 4.3 billion). All five tax and subsidy interventions had a 100% probability of being cost-saving. The sugar tax yielded the greatest health gains, followed by the salt tax, the fat tax, and the SSB tax. The subsidy alone did not result in a net health benefit due to possible adverse cross-price elasticity effects on the consumption of other foods. While not cost-effective on its own, the subsidy was a cost-effective addition to the overall tax package. The combined interventions had a 100% probability of being cost-saving across all scenarios [30].

Five diabetes prevention interventions evaluated in the United Kingdom (UK)—including SSB taxation, improved food access, workplace healthy eating promotion, a community-based men-only programme, and lifestyle intervention among high diabetes risk individuals—were all found to improve health outcomes and generate cost savings from a limited societal perspective that included employer productivity losses [25].

From a health system perspective in the UK, a diabetes prevention lifestyle intervention for high-risk adults produced 0.0003 to 0.0009 incremental QALYs and up to USD 1.68 in savings per person in the general population as outcomes were averaged across the entire population rather than only the intervention group. Cost-effectiveness increased with intervention intensity. The most cost-effective strategies were those targeting individuals with HbA1c > 6% or a high Finnish Diabetes Risk Score [24].

From a healthcare sector perspective in Australia, and using a model that accounted for competing risks, a 20% tax on SSBs was cost-saving across all scenarios. The intervention was estimated to generate 112,000 HALYs in men (95% UI 73,000 to 155,000) and 56,000 in women (95% UI 36,000 to 76,000), with healthcare savings of USD 567 million (95% UI USD 343 to 810 million). The tax was projected to generate USD 371 million in annual tax revenue. In all scenarios, the policy was likely to be cost-saving [32].

Osteoporosis screening for women aged 70-85 years in the UK resulted in a gain of 0.015 QALYs per patient (95% CI 0.007 to 0.023) and cost-savings of USD 454 (95% CI −919 to −122) [23]. Osteoporosis screening in Swedish men and women aged 50 and over—targeting those with osteoporotic fractures, secondary osteoporosis from glucocorticoid use or other high-risk factors—was estimated to save 14,993 QALYs and USD 973 million from a healthcare perspective [36].

Annual tobacco tax increases in New Zealand were estimated to result in 39,100 QALYs gained (95% CI 20,900 to 67,300) and USD 0.68 billion in healthcare cost savings (95% CI 0.37 to 1.15 billion) [37]. An air quality intervention in the UK—upgrading buses and heavy goods vehicles to EURO 6 standards—cost USD 10.37 million to implement and generated a one-off benefit of USD 5.43 million from reduced childhood asthma prevalence, along with annual benefits of USD 3.46 million linked to reductions in all-cause mortality, coronary events, low birthweight, and preterm births [26].

Screening for congenital Chagas disease among US women of childbearing age and newborns to infected mothers was cost-saving from a societal perspective, yielding lifetime savings of USD 193 per birth—equivalent to USD 773 million per annual birth cohort [16].

In the UK, behavioural interventions targeting diet, physical activity, and smoking—delivered through lay health workers, group wellbeing programmes, volunteering, and community activities—resulted in a total gain of 288 QALYs and cost savings of USD 6394/QALY. From a societal perspective—including productivity losses, social care costs due to adverse health outcomes, and criminal justice costs related to substance abuse—the interventions generated total public sector savings of USD 3.95 million [22].

#### 3.2.2. Cost-Effective Interventions (11 Studies)

Updated analyses of colorectal, breast, and cervical cancer screening programmes in the US accounted for health losses and lifetime healthcare costs from competing risks. From a healthcare sector perspective, this adjustment increased ICERs by USD 12,719 to 16,915/QALY. A colorectal cancer screening programme previously considered cost-saving was reclassified as cost-effective. At a threshold of USD 61,735/QALY, six of eight programmes remained cost-effective, and at USD 123,470/QALY, seven remained cost-effective [18].

Lung cancer screening for US adults aged 55 to 80 years with a 30 pack-year smoking history was evaluated using four independent and validated models from a healthcare sector perspective. All models concluded that the screening was cost-effective [19].

From a societal perspective, cervical cancer screening strategies for Norwegian women vaccinated against HPV were most cost-effective when a single lifetime screening was conducted for those vaccinated with the nonvalent vaccine, and two lifetime screenings were conducted for those who received the bivalent or quadrivalent vaccines [39].

From a societal perspective, a fall prevention intervention targeting UK adults aged 60 and older, tailored to individual fall risk, had a 93% probability of being cost-effective at a threshold of USD 28,736/QALY. It reduced health inequalities across socioeconomic status quartiles. While cost-ineffective among the oldest age groups, younger cohorts could cross-subsidise their older peers [28].

From a healthcare sector perspective in New Zealand, home assessment and modification to prevent falls among adults aged 65 years and older had an ICER of USD 4652/QALY (95% UI: cost-saving to USD 13,003/QALY). Targeting adults aged 75 years and older with prior injurious falls was cost-saving, with a median ICER of USD−74/QALY (95% UI: cost-saving to USD 3909/QALY) [40].

From a societal perspective in Sweden, a lifestyle intervention targeting weight reduction, increased physical activity, and healthier diet among pre-diabetic individuals had ICERs ranging from USD 5313/QALY (women, 30 years) to USD 12,775/QALY gained (men, 70 years). The probability of being cost-effective at a threshold of USD 88,303/QALY was high across all scenarios (85.0% to 91.1%) [41].

From a healthcare sector perspective in the UK, lifetime standard or high-intensity statin therapy was evaluated for individuals aged 40 and older, including those with and without preventive treatment recommendations under current guidelines. The analysis accounted for competing risks such as diabetes, cancer, and non-vascular deaths. Standard statin therapy was cost-effective across all groups. High-intensity therapy was cost-effective for individuals aged 40–70 with elevated cardiovascular disease (CVD) risk (QRISK3 ≥ 10%) or LDL ≥ 4.1 mmol/L, and for those aged 70 and older. Results were robust except among 40–49-year-olds with low 10-year CVD risk (5–10%) [27].

From a societal perspective in the US, pneumococcal vaccination with PCV20 compared to no vaccination was a dominant strategy for Black cohorts aged 50 and 65—improving QALYs while reducing costs. In non-Black cohorts, it was cost-effective but not cost-saving [17].

From a healthcare sector perspective in Canada, alcohol screening and brief intervention for adult primary care patients had an ICER of USD 8656/QALY [42].

From a healthcare sector perspective in the US, donor registry promotion had a cost per donor of USD 925,214 (95% CI 588,961 to 1.53 million), below the societal willingness-to-pay threshold of USD 1.38 million based on a QALY value of USD 127,440 [20].

From a public healthcare payer perspective in Poland, familial hypercholesterolemia (FH) screening across various target groups (children, young adults, or people after an acute coronary syndrome [ACS]) and using either clinical or genetic diagnostic methods was cost-effective (<USD 6859/QALY). The most cost-effective strategy involved screening ACS patients younger than 55 (men) and 65 (women), followed by cascade screening of their relatives [38].

#### 3.2.3. Cost-Ineffective Interventions (Two Studies)

From the public healthcare sector perspective in England, health checks for elderly individuals with intellectual disability were cost-ineffective, with a mean ICER of USD 147,699/QALY (95% CI 142,777 to 227,528), exceeding the cost-effectiveness threshold of USD 51,726/QALY [29].

From a patient perspective in the US, widespread genetic screening for FH, compared to family-based testing among cardiovascular disease-free young adults, was cost-ineffective, with ICERs ranging from USD 194,531 to 549,460/QALY depending on the screening start age [21].

### 3.3. Intervention Types

Regarding the content area of the interventions, among cost-saving interventions, diabetes and obesity prevention interventions were the most common (10 studies). Among the 11 studies analysing diabetes and obesity prevention interventions, one reported a cost-effective and not cost-saving result. Among cost-effective interventions, cancer screening and fall prevention interventions were the most common. Cost-ineffective interventions targeted rare diseases such as intellectual disability in the elderly and FH.

Considering the delivery approach of interventions, all regulatory interventions were cost-saving (12 studies), while most of the programme-based interventions were cost-effective (7 studies).

### 3.4. Analytical Features

#### 3.4.1. Perspectives

The most adopted perspective was the healthcare sector’s (15 studies), followed by societal (seven studies reporting from a societal perspective, three from a limited societal perspective), healthy system (one study), healthcare provider (one study), public healthcare payer (one study), and patient perspectives (one study).

#### 3.4.2. Discounting

Most studies applied an annual discount rate to both health outcomes and costs, between 1.5% and 4%. One study did not discount health outcomes [16], and two applied discounting only in sensitivity analyses, not in the base-case analysis [22,32].

#### 3.4.3. Competing Risks

Four studies modelled competing risks, all from a healthcare sector perspective. Two reported cost-saving regulatory interventions, while the other two showed cost-effective programme-based or clinical individual-level interventions.

#### 3.4.4. Morbidity Compression and Expansion Impacts

None of the included studies allowed for conclusions to be drawn regarding the theories of morbidity compression or expansion.

#### 3.4.5. Equity Analysis

Only six studies reported any equity analysis. The assessed equity impacts were heterogeneous: three studies showed benefits for ethnic minorities or low SES groups, one found no differential cost-effectiveness, one reported mixed results, and one identified unfavourable effects. For instance, pneumococcal vaccination among elderly in the United States was cost-saving in Black cohorts while only cost-effective in non-Black cohorts [17]. In the UK, a multifactorial fall prevention intervention targeting the elderly reduced health inequality across SES quartiles. Annual increases in tobacco taxes in New Zealand led to greater QALY gains among young Māori compared to non-Māori [37]. No differential cost-effectiveness was observed between Māori and non-Māori individuals for home assessment, and modification interventions aimed at fall prevention [40]. In the UK, the incremental net benefit of a diabetes prevention lifestyle intervention was smallest when targeting South Asian or low-SES groups [24]. Another study found that among diabetes prevention strategies, community-based and retail-level interventions preferentially benefited the most deprived quintiles, whereas individual-based approaches benefited higher SES groups [25].

### 3.5. Quality Appraisal

Most of the included studies fulfilled a majority of the Drummond 10-item quality criteria (see Appendix A). One study did not adjust the outcomes for differential timing [16], and one study did not perform any sensitivity analyses [26].

## 4. Discussion

To our knowledge, this is the first scoping review to examine the morbidity and net economic impact of health promotion and disease prevention interventions over a lifetime horizon. This approach allows for the inclusion of health outcomes and costs arising from competing risks during an individual’s remaining lifetime. However, only four studies included competing risks unrelated to the intervention in life years gained, and none reported whether the intervention led to morbidity compression or expansion.

This scoping review identified high-value health promotion and disease prevention interventions, as most studies reported cost-saving interventions, and over a third reported cost-effective results. Only a few studies described cost-ineffective interventions.

Clear thematic patterns emerged when comparing intervention types and cost-effectiveness results. Diabetes and obesity primary prevention interventions, osteoporosis screening, and regulatory measures—such as fiscal policies targeting tobacco, or vehicle regulation for air quality—were cost-saving. In contrast, programme-based or clinical individual-level interventions, including cancer screening, fall prevention, lifestyle counselling, and health checks, were cost-effective. Several explanations can be suggested for these trends that align with prior literature. Regulatory interventions tend to be more efficient than programme-based approaches [44]. Several factors explain this. First, they generally have lower implementation costs. programme-based or clinical interventions often require trained staff, physical infrastructure, and equipment, limiting scalability and cost-efficiency [45]. Second, regulations demand minimal ongoing investment, whereas programme-based and clinical interventions need sustained funding. Third, regulations affect the whole population and do not rely on individual-level participation. Nonetheless, when programme-based interventions were targeted to high-risk populations, such as individuals with prediabetes or a history of falls, they yielded more favourable cost-effectiveness outcomes. Cost-ineffective interventions targeted rare diseases.

Another pattern was observed between the perspective adopted and the cost-effectiveness result: interventions assessed from narrower perspectives were more frequently found to be cost-ineffective, whereas those evaluated from broader perspectives were more often cost-saving. This observed pattern reflects a well-established observation [46]. Broader perspectives capture spillover effects beyond the healthcare system. These include costs such as lost productivity due to patient or caregiver absenteeism, or criminal justice expenses related to substance use [13]. By including these broader consequences, societal perspectives often show more favourable cost-effectiveness outcomes [12,47].

Despite its importance, the equity impact of interventions was considered in only six of the 29 studies reviewed. Regulatory and community-based interventions demonstrated equity-enhancing effects. Programme-based interventions had mixed results, and clinical individual-level interventions showed the smallest health and costs benefits when targeting ethnic minorities or low SES groups. Several hypotheses can help explain the differing equity impacts observed across intervention types. Regulatory interventions apply universally and do not rely on individual engagement, which can be limited by factors such as health literacy or access to healthcare. With a population-wide reach, these interventions can have a greater impact on low-income groups, who are more sensitive to price changes induced by fiscal policies. Community-based interventions tend to promote equity by being locally tailored and co-designed, which enhances cultural relevance and accessibility. In contrast, programme-based and clinical individual-level interventions require active participation, which may be more feasible for higher SES groups.

Table 2 presents high-value promotion and prevention interventions for policy action—that is, cost-saving and cost-effective options. Interventions associated with a positive equity impact are shown in italics. Although a comparative ranking was not feasible due to the heterogeneity of reported economic outcomes, the table highlights interventions offering the strongest economic justification for public health investment. Cost-saving interventions reduce both healthcare system burden and overall expenditures, while cost-effective interventions provide good value for money based on QALYs gained. Prioritising these interventions can enhance population health and optimise healthcare spending. Diabetes and obesity prevention interventions, as well as regulatory interventions, emerge as high-return interventions. The high-value interventions identified in this review align with key government priorities in OECD countries, particularly in the areas of non-communicable disease (NCD) prevention and health equity. OECD public health strategies prioritise major NCD risk factors, including obesity, unhealthy diets, physical inactivity, environmental risks, and the harmful use of alcohol and tobacco [48]. In addition, the OECD emphasises the importance of addressing health-related inequalities as a core component of policy strategies “to promote inclusive growth and reduce social inequalities [49]”.

Our conclusions should be considered in light of several limitations. First, this is a scoping review, which limits the strength of the synthesis. The literature search was restricted to MEDLINE and only included studies published in English or French. Screening, data extraction, and quality appraisal were conducted by a single reviewer, introducing a potential source of bias despite efforts to ensure consistency. No risk of bias assessment was performed. Second, the included literature also has notable limitations. Few studies identified through the search strategy used a lifetime time horizon. This may be due to a focus on short-term clinical outcomes in empirical studies, which are not well-suited to capturing the long-term effects typical of promotion and prevention strategies. As a result, the available evidence mostly comes from modelling studies. Another key limitation is the variability in perspective choice. Only a minority of studies adopted comprehensive perspectives. This finding aligns with previous literature, which shows cross-country variation in recommended perspectives for economic evaluations [47]. These differences suggest the need for clearer guidance. Future health economic evaluation guidelines should better define the value and methodology of using a societal perspective. Because fully capturing societal costs can be difficult, a limited societal or health systems perspective may be a practical alternative [50]. Most notably, relatively few studies accounted for competing risks, which is essential to fully assess the long-term health and economic impacts of promotion and prevention interventions (see Table 3). The included studies varied in terms of modelling assumptions, methodology, and outcome metrics, limiting comparability.

Finally, as Neumann et al. reminded in a policy brief, “because health care resources are finite, […] it is useful to identify interventions that deliver the best value. […] While the achievement of cost savings through prevention is beneficial, it is important to keep in mind that the goal of prevention […] is to improve health [9]”, an intrinsically valuable good beyond its economic impact.

## 5. Conclusions

This review highlights high-value promotion and prevention strategies in ageing OECD populations that align with government priorities such as NCD prevention and health equity. Interventions targeting major risk factors—such as cardiovascular risk, environmental exposures, drug misuse, vaccination coverage, cancer screening, fall risk, and osteoporosis—were cost-saving or cost-effective, particularly when implemented through regulatory or targeted approaches, thereby supporting the case for upstream investment. Among the few studies that assessed equity impact, regulatory and community-based interventions were found to benefit disadvantaged groups.

However, methodological heterogeneity limited comparability across studies. Most cost-saving results came from analyses adopting broader perspectives. Only four studies accounted for competing risks—all from a healthcare sector perspective—and still reported cost-saving or cost-effective outcomes. Importantly, none of the studies assessed the impact on morbidity compression or expansion.

To inform sustainable and equitable health policy, future research should adopt standardised economic evaluation frameworks, including lifetime horizons, broader perspectives, and the integration of competing risks and equity impacts. These improvements will generate more robust, context-relevant evidence to guide long-term public health investment decisions.

## Figures and Tables

**Figure 1 ijerph-22-01161-f001:**
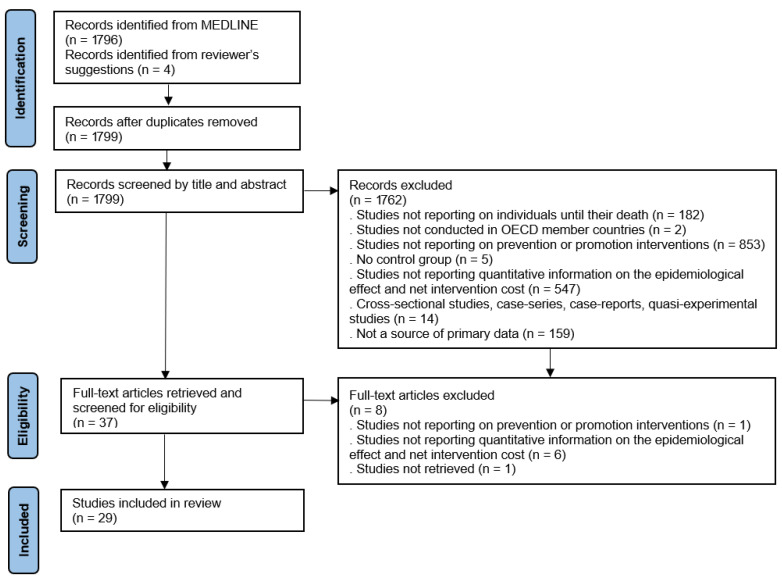
PRISMA flow diagram.

**Table 1 ijerph-22-01161-t001:** Summary of included studies by economic evaluation outcome.

Study Identification	Country	Intervention	Population	Perspective	Competing Risks	Equity Analysis
Cost-saving interventions						
Liu, 2022 [34]	Canada	Sweetened beverage tax	Adult population	Public healthcare payer	-	-
Olchanski, 2021 [14]	United States	Diabetes prevention	Pre-diabetes patients	Healthcare sector	-	-
Visram, 2020 [22]	United Kingdom	Integrated health and wellbeing service (1-to-1 interventions delivered by lay health workers, group wellbeing interventions, volunteering and community activities)	High-need population (veterans, socially isolated elderly, mental health issues, manual workers, LGBT groups)	Societal	-	-
Soreskog, 2020 [23]	United Kingdom	Osteoporosis screening	Women 70–85 years	Healthcare sector	-	-
Liu, 2020 [15]	United States	Menu calorie labelling	≥35 years	Societal	-	-
Ananthapavan, 2020 [31]	Australia	Obesity prevention	General population	Societal (limited)	-	-
Cleghorn, 2019 [35]	New Zealand	Cap on the size of SSBs	General population	Healthcare sector	-	-
Stillwaggon, 2018 [16]	United States	Congenital Chagas disease screening	Women of childbearing age; babies born to infected mothers	Societal	-	-
Jonsson, 2018 [36]	Sweden	Osteoporosis screening	≥50 years subject to osteoporosis treatment consideration (typical osteoporotic fracture, secondary osteoporosis due to glucocorticoid use, being at high risk due)	Healthcare sector	-	-
Cleghorn, 2018 [37]	New Zealand	Tobacco tax (10% annual increases)	General population	Healthcare sector	-	Greater QALY gains among Māori aged 20–65 years
Cobiac, 2017 [30,37]	Australia	Taxes on unhealthy food and drinks, and subsidies on healthy foods and drinks	General population	Healthcare sector	Modelled	-
Crino, 2017 [33]	Australia	Size cap and kilojoule reduction in SSBs	General population	Societal (limited)	-	-
Breeze, March 2017 [25]	United Kingdom	Diabetes prevention (five interventions)	General population, community-based, or high-risk individuals depending on the intervention	Societal (limited)	-	Community-based and retail policies preferentially benefited the most deprived quintile, whereas individual-based approaches benefited higher SES groups
Breeze, January 2017 [24]	United Kingdom	Diabetes prevention lifestyle intervention	High-risk adults for diabetes	Health system	-	The incremental net benefit was the smallest when targeting South Asian or low SES groups
Veerman, 2016 [32]	Australia	SSB tax	Adults	Healthcare sector	Modelled	-
Lomas, 2016 [26]	United Kingdom	Air quality intervention (upgrading pre-EURO 4 standards buses and heavy goods vehicles to EURO 6 standards)	General population	Healthcare sector	-	-
Cost-effective interventions						
Mihaylova, 2024 [27]	United Kingdom	Lifetime standard or high-intensity statin therapy	≥40 years, with and without statin therapy recommendation under current guidelines	Healthcare sector	Modelled	-
Altawalbeh, 2024 [17]	United States	Pneumococcal vaccination	50 and 65-year-olds	Societal	-	Cost-saving in the Black cohort, cost-effective in the non-Black cohort
Kwon, 2023 [28]	United Kingdom	Fall prevention	≥60 years	Societal	-	Health inequalities reduction by SES
Ratushnyak, 2019 [18]	United States	Breast, cervical, and colorectal cancer screening	Population eligible	Healthcare sector	Modelled	-
Criss, 2019 [19]	United States	Lung cancer screening	Current or former smokers aged 55 to 80 years; smoking history of at least 30 pack-years	Healthcare sector	-	-
Pelczarska, 2018 [38]	Poland	Familial hypercholesterolemia screening	Children, young adults, people after an acute coronary syndrome; relatives of the diagnosed-subjects	Public healthcare payer	-	-
Pedersen, 2018 [39]	Norway	Cervical cancer screening for women vaccinated against HPV infections	Women vaccinated against HPV infections	Societal	-	-
Wilson, 2017 [40]	New Zealand	Home safety assessment and modification	≥65 years	Healthcare sector	-	No differential cost-effectiveness by ethnicity (Māori vs. non-Māori)
Neumann, 2017 [41]	Sweden	Targeting weight reduction, increased physical activity and healthier diet	Pre-diabetic persons	Societal	-	-
Zur, 2016 [42]	Canada	Alcohol screening and brief intervention	Primary care adult patients	Healthcare sector	-	-
Razdan, 2016 [20]	United States	Donor registry promotion	General population	Healthcare sector	-	-
Cost-ineffective interventions						
Hendy, 2024 [21]	United States	Widespread genetic screening of familial hypercholesterolemia	Cardiovascular disease-free young adults	Patient	-	-
Bauer, 2019 [29]	England	Health checks for the elderly with intellectual disability	≥40 years, with intellectual disability	Healthcare provider	-	-

HPV; Human Papillomavirus, LGBT; lesbian, gay, bisexual, transgender groups, SES; socio-economic status, SSB; sugar-sweetened beverage.

**Table 2 ijerph-22-01161-t002:** High-value promotion and prevention interventions: cost-saving and cost-effective options for policy action.

Cost-Saving Interventions	Cost-Effective Interventions
*Community-based diabetes and obesity prevention*	*Fall prevention among 65 years and older*
*Tobacco taxation*	Statin therapy as primary and secondary prevention among 40 years and older
*Pneumococcal vaccination in the elderly*	Familial hypercholesterolemia screening
Diabetes and obesity prevention	Cancer screening (colorectal, lung, breast, cervical)
Air quality intervention	Alcohol screening
Osteoporosis screening	Donor registry promotion
Fall prevention among 75 years and older	
Wellbeing services (1-to-1 health behaviour interventions by lay health workers, group wellbeing interventions, volunteering and community activities)	
Congenital Chagas disease screening in the United States	

Interventions with a positive equity impact are shown in italics.

**Table 3 ijerph-22-01161-t003:** Future research priorities around the economic evaluation of health promotion and disease prevention.

◦ Generate evidence on whether promotion and prevention interventions contribute to morbidity compression or expansion
◦ Recommend the incorporation of competing risks
◦ Encourage high-quality by recommending the adoption of lifetime horizons, and limited societal or health systems’ perspectives

## Data Availability

The original contributions presented in this study are included in the article. Further inquiries can be directed to the corresponding author.

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
