# Peer review of "A Scoping Review on the Economic Impacts of Healthy Ageing Promotion and Disease Prevention in OECD Member Countries"

_ijerph, 2025, doi:10.3390/ijerph22081161_

Round 1
Reviewer 1 Report
Comments and Suggestions for Authors
This study used a scoping review to evaluate the lifetime economic impact of health promotion or primary and secondary prevention interventions in OECD countries.
The manuscript was well-constructed and well-written.
The inclusion criteria for the time horizon of a lifetime excluded many studies. The results (or lack thereof) of this review regarding the impact of interventions on morbidity compression or expansion, as well as equity, were not surprising.
Methods
“A single reviewer (the first author) conducted the data extraction process.” Suggest adding this as a limitation.
Did you review the references in recent papers to identify other relevant studies that may need to be included?
Here are a few papers that I did not see in your references. They may satisfy the inclusion criteria for this study.
Breeze PR, Thomas C, Squires H, et al. The impact of type 2 diabetes prevention programmes based on risk-identification and lifestyle intervention intensity strategies: a cost-effectiveness analysis. Diabet Med 2017;34:632–640
Cobiac LJ, Tam K, Veerman L, Blakely T. Taxes and subsidies for improving diet and population health in Australia: a cost-effectiveness modelling study. PLoS Med 2017;14:e1002232
Veerman JL, Sacks G, Antonopoulos N, Martin J. The impact of a tax on sugar-sweetened beverages on health and health care costs: a modelling study. PLoS One 2016;11:e0151460
Neumann A, Lindholm L, Norberg M, Schoffer O, Klug SJ, Norström F. The cost-effectiveness of interventions targeting lifestyle change for the prevention of diabetes in a Swedish primary care and community based prevention program. Eur J Health Econ 2017;18:905–919
Reviewer 2 Report
Comments and Suggestions for Authors
Dear authors,
I was pleased to read your paper.
In this regard, I have a few suggestions.
The manuscript states that data extraction was completed on May 6. If the analysis and drafting occurred within less than a month, this may raise concerns regarding the depth of data handling and synthesis. Especially with just two authors.
A serious methodological shortcoming of this paper is the participation of only one researcher in data extraction. This may imply serious bias.
Please note the absence of other languages in the limitations.
Another characteristic of this paper is the text overload. Try to write the text in such a way that readers could, for example, see in paragraphs 150 to 161 which references come from which country. Try to introduce subheadings to make it easier to follow the text.
Below Table 1, the entire text is overloaded with numbers. It boils down more to recounting studies than to critically evaluating and comparing them.
Consider some of these:
- Results
3.1 Overview of Included Studies
.....
Distribution by country (list only the most important, leave the rest to the table)
3.2 Economic Impact Categories
3.2.1 Cost-saving interventions - with text summary and table references
3.2.2 Cost-effective interventions
3.2.3 Not cost-effective interventions
3.3 Intervention Types
group by type of intervention: diabetes, obesity, screening, etc.
for example, "Among cost-saving interventions, obesity prevention and beverage taxation were most common."
3.4 Analytical Features
Perspective (healthcare sector vs. societal)
Discounting
Morbidity Compression (none addressed)
Equity analysis (short)
In the Discussion, use shorter sentences with the verb as close to the beginning as possible. Introduce some other studies to explain the discussion points.
Table 2 is just introduced and left without further explanation.
The restrictions are too long.
As well as the conclusion with the lack of focus.
I recommend the bullet-point list as a policy brief.
I think that these suggestions are feasible and that you can implement them quickly. After which the article would be ready for publication.
Best
Reviewer 3 Report
Comments and Suggestions for Authors
Review for manuscript ijerph-3696176 `A scoping review on the economic impacts of healthy ageing promotion and disease prevention in OECD member countries`
Thank you for the opportunity to review manuscript ijerph-3696176.
The manuscript cover a topic of examining the economic value of health promotion and disease prevention interventions within ageing populations in OECD countries. It successfully synthesizes a wide range of economic evaluations using a lifetime horizon and offers useful policy-relevant insights. However, the manuscript in its current form requires substantial revision to meet scholarly standards for publication, particularly in methodology reporting, data synthesis.
General comments:
Methods and materials section:
Page 3, lines 100–101
Although the authors acknowledged that the search was restricted only to MEDLINE via Ovid, this limits the comprehensiveness of the review. Other databases for economic and health policy literature such as EconLit, Embase, PubMed and others were not included. Therefore, I recommendation to include at least one economics-focused database and one multidisciplinary database to ensure broader coverage of the literature and non-clinical evaluations.
Page 3, line 123
It was noted that the selection and data extraction were performed by a single reviewer, which is prone to selection and extraction bias. This weakens the reliability of the synthesis. Therefore, I recommendation to employ a second reviewer to validate selection and extraction or, if not possible, include an explanation and discussion of potential bias in the limitations section (line 406).
Page 3, lines 97–98 and Page 6, lines 409–410
The authors acknowledged that no formal quality appraisal or risk of bias assessment using standardized checklists was conducted and provided only a brief explanation. While this is sometimes acceptable in scoping reviews, consider applying a standardized tool, like the CHEERS 2022 checklist or the Drummond criteria, given that the study includes economic evaluations. These tools are specifically designed to assess the reporting quality and methodological rigor of economic evaluations. Even if a full critical appraisal is beyond the scope of the review, a brief structured assessment could enhance the transparency and interpretability of findings, especially for readers interested in the reliability of economic evidence. Therefore, I recommendation either perform a basic quality check or provide stronger justification for omitting it. Authors at least can discuss the potential variation in study quality as a limitation.
Page 3, lines 130–135
It is unclear whether specific criteria for identifying morbidity compression or expansion were applied during data extraction. Please clarify whether studies that did not disaggregate morbidity data from longevity outcomes were excluded, and if not, how such studies were handled in the synthesis.
Results and Discussion:
Page 4 Lines 356
Some statements in the discussion are not linked with specific reference/citation. For example, entire part - lines 384-405 – has only two citations. Particularly, in this sentence “…well-documented effect” was mentioned without citations: “Regarding the observed trends between restrictive perspectives and cost-ineffective results, and more comprehensive perspectives and cost-saving results, this reflects a well-documented effect.”
Result and discussion sections present primarily descriptive summaries of included studies but lacks deeper analytical or thematic synthesis. A more rigorous narrative synthesis is required to draw meaningful conclusions. Discussion section needs more elaboration and integration with current literature on the topic.
Also, findings are not clearly linked to specific policy actions or recommendations. For instance, what should OECD governments prioritize in prevention portfolios based on the evidence? Therefore, I recommendation to enhance the policy relevance by discussing how the cost-saving/cost-effective interventions align with government decision-making priorities (e.g., in universal health coverage, health taxes, or NCD strategies).
Conclusion
Page 7, lines 456–459
Authors mentioned equity impact, I think it deserves more detailed discussion. Please consider to include a table or summary of which studies addressed equity and what dimensions (SES, ethnicity) were considered.
The conclusion reiterates background information rather than synthesizing findings or highlighting policy/practice implications. It should clearly summarize what was learned, which interventions/studies showed the greatest impact, and where further research is needed.
Minor comments:
Page 1-2, lines 33–41
Looks like there is a confusion between COVID-related expenditures and planned investments. Please clarify with reference to OECD health expenditure reports.
Page 7, lines 449
Proofread for minor grammatical mistakes: “However. The only two …”
Round 2
Reviewer 3 Report
Comments and Suggestions for Authors
Thank you for addressing my comments. Good luck.